# The Chemical Element Composition of Turmeric Grown in Soil–Climate Conditions of Tashkent Region, Uzbekistan

**DOI:** 10.3390/plants10071426

**Published:** 2021-07-12

**Authors:** Dilfuza Jabborova, Ravish Choudhary, Rohini Karunakaran, Sezai Ercisli, Jyoti Ahlawat, Khurshid Sulaymanov, Abdulahat Azimov, Zafarjon Jabbarov

**Affiliations:** 1Laboratory of Medicinal Plants Genetics and Biotechnology, Institute of Genetics and Plant Experimental Biology, Uzbekistan Academy of Sciences, Tashkent Region, Kibray 111208, Uzbekistan; dilfuzajabborova@yahoo.com (D.J.); ksulaymanov@yahoo.com (K.S.); ahat2021@yahoo.com (A.A.); 2Division of Microbiology, ICAR-Indian Agricultural Research Institute, Pusa, New Delhi 110012, India; 3Division of Seed Science and Technology, ICAR-Indian Agricultural Research Institute, New Delhi 110012, India; ravianu1110@gmail.com; 4Unit of Biochemistry, Faculty of Medicine, AIMST University, Semeling, Bedong 8100, Malaysia; 5Department of Horticulture, Agricultural Faculty, Ataturk University, Erzurum 252240, Turkey; sercisli@gmail.com; 6Department of Biotechnology, GD Goenka University, Gurugram, Haryana 122102, India; j.10dec@gmail.com; 7Faculty of Biology, National University of Uzbekistan, Tashkent 100174, Uzbekistan; zafarjonjabbarov@gmail.com

**Keywords:** turmeric, macroelements, microelements, mineral fertilisers, soil nutrients, soil enzymes

## Abstract

A mineral fertiliser has positive effects in improving turmeric nutrients, soil enzymes and soil properties. The aim of this research was to study the effect of mineral fertilisers on the content of mineral elements in turmeric rhizome, soil enzymes activity and soil properties in the Tashkent Region, Uzbekistan. For the first time in Uzbekistan, the turmeric rhizome was cultivated to study the mineral elements present in the rhizome. A microplot experiment was conducted with four treatments including T1 (Control), T2 (N_75_P_50_K_50_ kg/ha), T3 (N_125_P_100_K_100_ kg/ha) and T4 (N_100_P_75_K_75_ + B_3_Zn_6_Fe_6_ kg/ha) and turmeric rhizome, which were collected for observation along with the soil samples. The analyses indicated that the NPK + BZnFe (100:75:75:3:6:6 kg/ha) treatment significantly improved minerals such as K, Ca, P, Mg and Na contents rhizome as compared to the control without fertiliser. Likewise, the maximum quantity of micronutrient content viz., Fe, Mn, Zn, Cu, Cr and Si was also recorded in turmeric rhizome treated with NPK + BZnFe (125:100:100:3:6:6 kg/ha). It showed an increase in these micronutrients in the rhizome compared to the control, followed by a low rate of NPK (75:50:50 kg/ha). The highest content in terms of total N, P, K content, humus, active phosphorus, potassium, and enzymes activity was also observed in soil with the treatment of mineral fertiliser viz., NPK + BznFe (100:75:75:3:6:6 kg/ha), which enhanced soil nutrient and enzyme activity. The NPK + BznFe (100:75:75:3:6:6 kg/ha) treatment significantly increased the active N content by 40%, total P content by 38% and total K content by 22% in comparison to the control without mineral fertiliser. Overall, it was found that NPK + BznFe (100:75:75:3:6:6 kg/ha) was significantly valuable for enhancing the total nitrogen, phosphorus, and potassium levels in the soil compared to control, which is useful for improving soil health in terms of soil enzyme and soil nutrients. Additionally, the micronutrients in turmeric rhizome were significantly enhanced when using this combination of fertiliser applications [NPK + BznFe (100:75:75:3:6:6 kg/ha)]. Therefore, this present study revealed that the NPK+BznFe (100:75:75:3:6:6 kg/ha) could produce the most significant yield of high-quality turmeric plants and improve soil properties in Uzbek soil–climate conditions.

## 1. Introduction

Herbal plants are an important source of traditional medicine and play a key role in global health [1,2,3,4,5]. Turmeric (*Curcuma longa* L.) belongs to the family *Zingiberaceae*, which includes rhizomatous herbaceous perennial herbs with heights of 60 to 90 cm; there are more than 40 commercially cultivated species of *Curcuma longa* [6]. It is a tropical South Asian native crop which is widely cultivated in the tropical and subtropical regions of the world. The yellow-coloured turmeric powder, mainly used as spices or traditional Ayurvedic medicine in India, is comprised of half-boiled and dried rhizomes. It is extensively used as a stimulant, carminative, aspirant, cordial, astringent, diuretic, martinet and anti-inflammatory, anticancer, chronic anterior uveitis and pancreatitis, cardioprotective [7,8], hypoglycaemic [9], antiamyloidogenic [10], antifungal [11], parasitical [12] and antioxidant treatment [13,14]. Its aroma and colour come from the essential oils and curcumin present in turmeric [15]. Additionally, curcumin plays a vital role in inducing apoptosis and antiangiogenic activity [16,17].

As *Curcuma* contains curcumin, which makes it a natural food additive and food colourant, and has medicinal properties, such as anti-cancer and antiviral properties, global demand for it is likely to increase, especially in Western countries. Curcumin has a long scientific history dating back to ancient times and has attracted modern research interest for the past four decades. It is rich in several essential minerals such as Fe, Ca, Mg, and P and vitamin A and widely used in drinks and supplements [18].

The fertility of soil is maintained by adding extra fertilisers and different kinds of manure and macronutrients from chemical fertilisers [19]. Soil property and crop yield can be improved using main elements such as N, P, and K and secondary nutrients such as Mg in either organic or inorganic form. Mg plays an essential role in many biochemical and physiological processes in plants [20]. Foliar spray with a higher concentration of MgSO_4_ increases yields of turmeric rhizome [21]. The combination of different fertilisers influences soil fertility, the soil’s chemical properties, and crop yield and quality. Mineral fertiliser, soil pH and physiological properties were the most necessary for the higher mineral content of turmeric. Earlier, it was found that the local climatic and edaphic factors are essential to realising a high yield and high quality in turmeric cultivation [22,23]. Hossain and Yukio [24] also reported that grey soil containing large amounts of K and Ca led to the highest amount of K, Ca and Mg contents in turmeric rhizomes. 

In the Tashkent region of Uzbekistan, the soil is impoverished in terms of nutrients and minerals as turmeric is a long-duration crop and needs a high amount of additional nutrients and minerals in the soil for higher yield. Naturally, there are not enough nutrients in the soil of the Tashkent region for good quality turmeric production, as the soil of the experimental site was deficient in essential nutrients, resulting in a decrease in turmeric growth and yield. Therefore, keeping these views in mind, the present study has been undertaken to determine mineral fertilisers’ effects on the analysis of the mineral nutrients in turmeric rhizome and soil enzyme properties.

## 2. Results and Discussion

### 2.1. Impact of Mineral Fertilisers on Turmeric Rhizome Nutrients

The results showed that the application of mineral fertiliser treatments such as NPK (125:100:100 kg/ha) and NPK + BZnFe (100:75:75:3:6:6 kg/ha) increased the macroelement content in the turmeric rhizome (Table 1). The NPK (125:100:100 kg/ha) significantly increased rhizome K, Ca P and Mg content compared to control without fertiliser. Similarly, the NPK + BZnFe (100:75:75:3:6:6 kg/ha) treatment with the highest amount of fertiliser was also significantly increased macroelement content compared to control. However, NPK + BZnFe (125:100:100:3:6:6 kg/ha) treatment showed a significant enhancement in rhizome K, Ca, P, Mg and Na contents over the control (Table 1). Several researchers have been reported the macro–micro elements in different plants [25,26,27]. Mineral nutrients such as N, P, K play an essential role in plant growth, development, yield and productivity [27,28]. According to Adekiya et al. [18], it was suggested that rhizome growth and yield were increased with the supply of high concentrations of nutrients and low C:N ratio, which resulted in an increase in decomposition with the nutrient release. A similar study was reported by Ihenacho et al. [29] in their study on turmeric, as the deficiency of essential nutrients in the soil of the experimental site resulted in a lower yield and decrease in turmeric growth [30]. The nitrogen content of NPK formed proteins and chlorophyll content in plants which induced leaf growth. At the same time, phosphorus is necessary for root development, cell division, multiplication, and energy reactions, and potash worked for stem development, cell division, carbohydrates formation and translocation in plants, especially turmeric; it is useful for rhizome development [31]. It was reported that other macroelements such as Mg, used for nitrogen uptake enhancement in plants, as the application of Mg increased the root growth by translocation and absorbed the other macronutrients in the soil, increasing rhizome growth and yield of turmeric [32]. The literature showed no studies have so far been undertaken on the determination of mineral nutrients in turmeric cultivated in Uzbekistan.

The results on microelements found in turmeric rhizome are presented in Table 2. The microelements in turmeric rhizome were significantly increased by the NPK (125:100:100 kg/ha) and NPK + BZnFe (100:75:75:3:6:6 kg/ha) treatments (Table 2). Data regarding rhizome microelement content showed that NPK (125:100:100 kg/ha) significantly enhanced the contents of micronutrients such as Fe, Mn, Zn, Cu, Cr and Si in turmeric rhizome as compared to control. A maximum quantity of micronutrient content in turmeric rhizome was recorded with NPK + BZnFe (125:100:100:3:6:6 kg/ha), which increased Fe, Mn, Zn, Cu, Cr and Si contents in rhizome over the control and low rate of NPK (75:50:50 kg/ha). Nutrient analysis by Olubunmi et al. [33] of other tuber crops such as ginger indicated their richness in calcium, magnesium, sodium, potassium, phosphorous, manganese, iron, zinc, and copper. Poultry manure alone or in combination with Mg fertiliser resulted in increasing other several nutrients viz., Na, K, Mg, Ca, Fe, and vitamin contents in turmeric rhizome as compared with NPK with or without Mg fertiliser due to positive effect on soil and plants [34].

Data regarding the ultra-microelement content in turmeric rhizome showed a non-significant difference between all the treatments. For most ultra-micronutrients (Li, Be, V, Co, Ni, Ga, Ge, Nb, Ag, Cd, Sn, Sb, Cs and W content), control treatment showed higher values of rhizome nutrients than other fertiliser treatments (Table 3), while micronutrients such as In, Ta and Re were absent in turmeric rhizome in all the treatments and control. Similarly, the soil enriched with additional sulphur and magnesium resulted in the higher rhizome yields in the kacholam crop [35]. In the same way, Fe application also increased the total yield in the fenugreek crop [36]. The effects of Zn, Fe and B were also observed to be beneficial for total rhizome growth and yield in tuber crops such asginger [37] and turmeric [38].

### 2.2. Impact of Mineral Fertilisers on Soil Properties

Results for soil mechanical composition are presented in Table 4. The data show that treatments T3 (N125P100K100 kg/ha) and T4 (N100P75K175 + B3Zn6Fe6 kg/ha) increased the mechanical composition of the soil, whereas treatment T4, including macro and micronutrients (N100P75K75 + B3Zn6Fe6 kg/ ha) with the highest amount of fertiliser, significantly increased soil mechanical particles (1.0–0.25 mm, 0.1–0.05 mm, 0.05–0.01 mm) as compared to control without fertiliser.

Mineral fertilization with nitrogen, potassium, phosphorus, and micronutrients (B, Zn, and Fe) positively affected active P, K, N content, total P and K content, and organic matter in the soil (Table 5). The highest values of total P, K and N content, organic matter, active phosphorus, and potassium were observed in soil with mineral fertiliser treatments. NPK (125:100:100 kg/ha) and NPK + BZnFe (100:75:75:3:6:6 kg/ha) improved essential nutrients in soil as compared to control and NPK (75:50:50 kg/ha) treatments. The NPK + BZnFe (100:75:75:3:6:6 kg/ha) treatment significantly increased active P content by 40%, total P content by 38% and total K content by 22% in comparison to the control without mineral fertiliser. Several other authors also reported the nutrient contents in the soil before and after plant cultivation [39,40]. Some other studies about poultry manure showed limitations so that NPK alone or with a combination of other nutrient fertilisers could increase the essential macronutrients in soil [41]. Dinesh et al. [42] reported the chemical nutrient management in soil and found enhancement of total N content in soil containing rainfed ginger (*Zingiber officinale* Rosc.) tuber crops. Similarly, the N, P and K content in soil was also increased using NPK application (100: 60: 60 kg/h) [43].

As a result of the experiments, there was a significant change in chlorine and sulphate ions. The amount of chlorine ion was found to be 0.20 mg/eq with control, while the NPK + BZnFe application (100:75:75:3:6:6 kg/ha) showed 0.13 mg/eq in the soil (Table 6). The NPK + BZnFe (100:75:75:3:6:6 kg/ha) treatment enhanced Ca and Mg content significantly compared to all other treatments. Overall, the results showed that the NPK + BZnFe (100:75:75:3:6:6 kg/ha) significantly improved soil properties.

### 2.3. Impact of Mineral Fertilisers on Soil Enzyme Activity

The data show that mineral fertilisers increased urease activity in the soil (Figure 1). The urease activity was significantly enhanced by the treatment of NPK (125:100:100 kg/ha). The maximum urease activity was recorded using NPK + BZnFe (100:75:75:3:6:6 kg/h) compared to control and other treatments. NPK fertiliser and other microelements increased the N, P, and K concentrations in the soil. The amount of organic Ca, Mg, and pH in the soil was increased due to excellent decomposition and resulted in the highest release of organic matter and other nutrients inside the soil. The soil pH increased due to the presence of base cations which are released upon microbial decarboxylation [44].

In the present study, the data indicated that the increased fertiliser combinations of the NPK (125:100:100 kg/ha) and NPK + BZnFe applications (100:75:75:3:6:6 kg/ha) significantly enhanced invertase activity in the soil (Figure 2). The application of NPK (125:100:100 kg/ha) significantly increased invertase activity by 50% compared to control. In contrast, the combination of macro and micronutrients, NPK + BZnFe (100:75:75:3:6:6 kg/ha), significantly enhanced the invertase activity by 58% more than in control without fertiliser.

The data in Figure 3 show that mineral fertilisers increased the catalase activity of the soil. The increases in catalase activity reached a maximum with a higher amount of NPK (125:100:100 kg/ha) and NPK + BZnFe (100:75:75:3:6:6 kg/ha) treatments as compared to the control. However, NPK + BZnFe (100:75:75:3:6:6 kg/ha) showed a significant increase in catalase activity over the control and other treatments—T-2 and T-3.

Overall the results of the present study showed that the NPK + BZnFe application (100:75:75:3:6:6 kg/ha) significantly increased all three enzymes such as urease (Figure 1) invertase (Figure 2), and catalase activity (Figure 3) in soil. Similar findings were confirmed earlier by Srinivasan et al. [45] and reported that NPK (75: 50: 50 kg/h) increased the urease activity by 27% in the soil. In this context, Singh et al. [46] and Allison et al. [47] observed the enzymatic activities in soil, which was enhanced by applying different mineral fertilisers. The literature showed no studies so far that have reported on the determination of soil enzyme activity during turmeric cultivation in Uzbekistan.

For the first time in Uzbekistan, the content of mineral elements in cultivated turmeric rhizome was studied. This study clearly explained that the NPK + BZnFe (100:75:75:3:6:6 kg/ha) significantly increased the macro and micronutrient contents in turmeric rhizome. The combined application of the NPK + BZnFe (100:75:75:3:6:6 kg/ha) also significantly improved the essential nutrients and enzymatic activities in the soil. These results suggested that the NPK + BZnFe (100:75:75:3:6:6 kg/ha) can produce the most significant yield with high quality of turmeric rhizome improving soil properties in Uzbek soil–climate conditions.

## 3. Materials and Methods

### 3.1. Experimental Design

Turmeric (*Curcuma longa*) rhizome was used for this present study. A microplot experiment was conducted to study the effect of mineral fertilisers on mineral nutrients in turmeric rhizome and soil properties. The experiment was carried out in randomized block design with three replications at the Institute of Genetics and Plant Experimental Biology, Kibray, Tashkent Region, Uzbekistan. Experimental treatments included:T1—ControlT2—N75P50K50kg/haT3—N125P100K100kg/haT4—N100P75K75+B3Zn6Fe6kg/ha

Rhizomes were sown on 18 March 2019, and harvesting was performed on 18 November 2019.

### 3.2. Measurement of Plant Nutrients

Turmeric rhizomes were harvested after 240 days of cultivation. These harvested rhizome samples were prepared for analysis. Analysis was carried out in a special autoclave under the influence of hydrogen peroxide (H_2_O_2_) and nitric acid (HNO_3_) as disintegrating reagents for 6 h using a special microwave oven until the plant samples were converted into atomic elements. Sample volumes were accurately measured, and then 2.0% HNO_3_ was added. The analysis was carried out on an optical emission spectrometer with inductively coupled argon plasma (2100DV; USA) [48].

### 3.3. Analysis of Soil Nutrient

Soil samples were collected randomly from a microplot of the experimental site in three replicates. To determine the soil properties before experimenting, soil samples were taken. The mechanical composition of the soil was determined by modified Kachinsky’s method [49]. Mobile compounds of phosphorus and potassium were determined by the Michigan method modified by CINAO [50], while the total phosphorus and potassium contents were determined using another modified method [51]. The total nitrogen content was observed according to the method of Soils [52]. The salinity level of soil was also determined by the water extraction method [53]. Analysis of soil properties is shown in Table 7,Table 8,Table 9.

### 3.4. Analysis of Soil Enzymes

Urease activity of soil was assayed using the method by Xaziev [54]. Total 2.5 g of soil sample were mixed with 0.5 mL of toluene for 15 min. Then, 2.5 mL of 10% urea and 5 mL citrate buffer were added and incubated at 38 °C in an incubator for 24 h. Then, this working mixture was diluted with double distilled water and filtered. Then, 4.0 mL of sodium phenate with 3.0 mL of sodium hypochlorite were added into 1.0 mL of filtrate and diluted with 50 mL and kept for 20 min at room temperature. Urease activity (mg NH_4_/g) of soil was measured at wavelengths of 578 nm using a spectrophotometer. Invertase and catalase activity of soil were also assayed using the method described by Xaziev [54]. In the invertase activity, a total of 5.0 g of dried soil was used for the experiment. In this dried soil, 15.0 mL of sucrose solution (8.0%) with 5.0 mL of double distilled water were added. Then, finally, in the end, five drops of toluene were added and keep this soil mixed solution for 24 h incubation at 37 °C.

Then, incubated soil solution was centrifuged at 4000 rpm for 5 min. Then, 1.0 mL aliquot collected from the vial after centrifugation and was transferred into a volumetric flask containing 3.0 mL of 3.5-dinitrosalicylic acid and kept on the hotplate for 5 min. Finally, the solution was kept for cooling up to room temperature. Then, quantification of glucose content was performed using the colorimetric method at the wavelength of 508 nm on a spectrophotometer. Invertase activity was expressed as μg glucose·g^−1^ soil·h^−1^. For catalase activity of the soil, a total of 2.0 g of thoroughly air-dried soil was mixed with 5.0 mL of H_2_O_2_ and 40.0 mL of double-distilled water. The mixture of this soil with this solution was shaken for 20 min at 150 rpm. The remaining hydrogen peroxide (H_2_O_2_) was stabilized using 5.0 mL of sulfuric acid (1.5 M H_2_SO_4_) followed by centrifuging at 4000 rpm for 5 min. Then, the supernatant was used for titration with 0.05 M KMnO4. Catalase activity (mL KMnO_4_ g^−1^ soil h^−1^) of soil was measured at a wavelength of 480 nm using a spectrophotometer.

### 3.5. Statistical Analyses

All the experiments were replicated three times, and the mean values were considered. The data were statistically analysed by one-way analysis of variance (ANOVA) and multiple comparisons of HSD employing Tukey’s test with Stat View Software (SAS Institute, Cary, NC, USA, 1998). The magnitude of the *p*-value determined the significance of various treatments on plant macro, micro, ultra-micronutrients, and soil nutrients (*p* < 0.05, 0.01 and 0.001).

## Figures and Tables

**Figure 1 plants-10-01426-f001:**
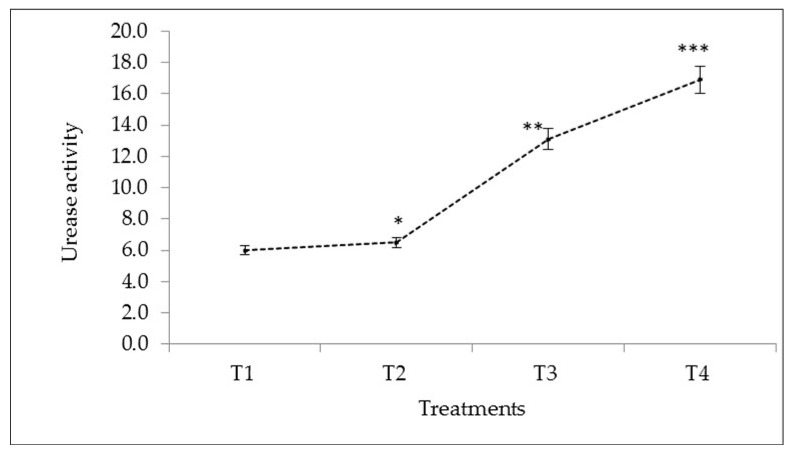
The effect of mineral fertilisers on urease activity of irrigated soil in Kibray district, Tashkent Region, Uzbekistan. (*p* values showing significantly different at *p* < 0.05 *, *p* < 0.01 **, *p* < 0.001 *** as compared to control (T1)). Bars represent ± SD values.

**Figure 2 plants-10-01426-f002:**
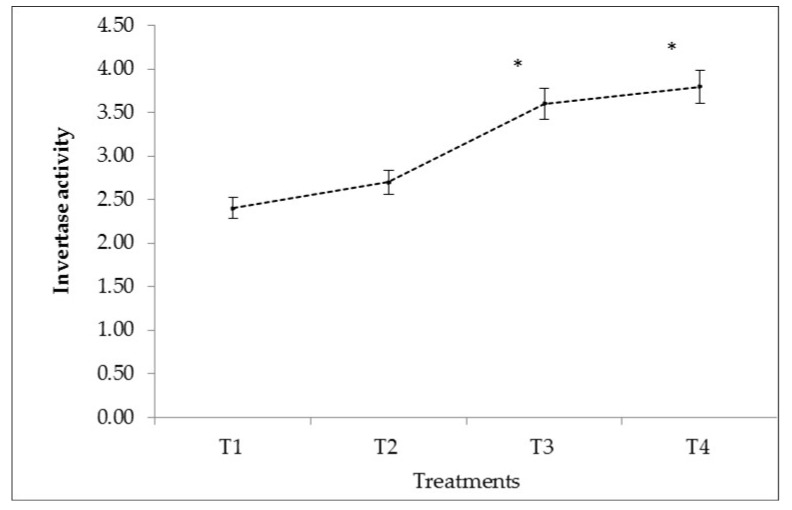
The effect of mineral fertilisers on invertase activity of irrigated soil in Kibray district, Tashkent Region, Uzbekistan. (*p* values showing significantly different at *p* < 0.05 * as compared to control (T1)). Bars represens ± SD values.

**Figure 3 plants-10-01426-f003:**
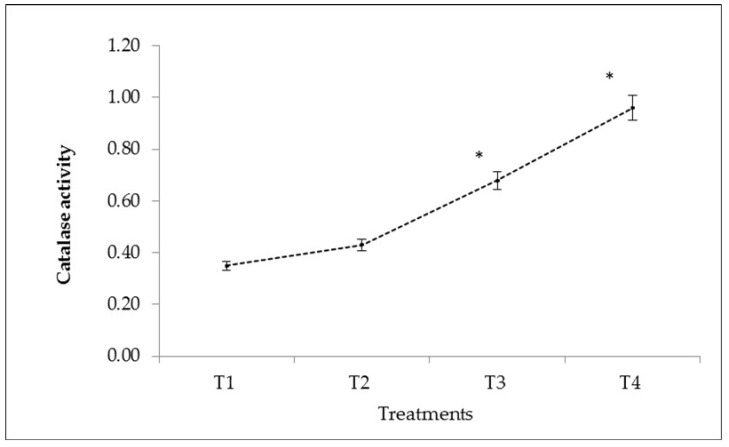
The effect of mineral fertilisers on catalase activity of irrigated soil in Kibray district, Tashkent Region, Uzbekistan. (*p* values showing significantly different at *p* < 0.05 * as compared to control (T1)). Bars represent ± SD values.

**Table 1 plants-10-01426-t001:** The impact of mineral fertilisers on macroelement content of turmeric rhizome grown in soil–climate conditions of Tashkent Region.

Macroelements (mg/kg)	Treatments			
	Control	N_75_P_50_K_50_	N_125_P_100_K_100_	N_100_P_75_K_175_ + B_3_Zn_6_Fe_6_
K	14223.46	18536.65	24812.78 *	20541.87
Ca	1958.57	2236.11	8865.55	13261.19 *
P	5916.50	6649.30	7075.49	6795.69 *
Mg	4544.58	5516.49	6733.86	7598.70 *
Na	194.75	246.01	300.72	823.38 *

* All the values are the average of three replicates (*n* = 3), and the mean with the same letter (superscript) in the columns are not significantly different (*p* < 0.05)—(Tukey’s test).

**Table 2 plants-10-01426-t002:** The impact of mineral fertilisers on microelement content of turmeric rhizome grown in soil–climate conditions of Tashkent Region.

Microelements (mg/kg)	Treatments			
	Control	N_75_P_50_K_50_	N_125_P_100_K_100_	N_100_P_75_K_175_ + B_3_Zn_6_Fe_6_
Fe	79.32	95.23	111.81	217.02 *
Mn	6.67	7.12	8.37	11.65 *
Zn	2.53	5.65	6.57	9.89 *
Cu	1.80	1.92	1.95 *	1.95 *
Cr	0.95	1.15 *	1.15 *	1.15 *
Mo	0.062	0.064 *	0.064 *	0.064 *
Si	568.54	678.53	684.42	699.43 *

* All the values are the average of three replicates (*n* = 3), and the mean with the same letter (superscript) in the columns are not significantly different (*p* < 0.05)—(Tukey’s test).

**Table 3 plants-10-01426-t003:** The impact of mineral fertilisers on ultra-microelement content of turmeric rhizome grown in soil–climate conditions of Tashkent Region.

Treatments	Treatments			
	Control	N_75_P_50_K_50_	N_125_P_100_K_100_	N_100_P_75_K_175_ + B_3_Zn_6_Fe_6_
Li	0.190 *	0.190 *	0.190 *	0.190 *
Be	0.008	0.008	0.009 *	0.009 *
V	0.171 *	0.171 *	0.171 *	0.171 *
Co	0.048 *	0.036	0.042	0.044
Ni	0.416*	0.416 *	0.416 *	0.416 *
Ga	0.222 *	0.222 *	0.222 *	0.222 *
Ge	0.001 *	0.001 *	0.001 *	0.001 *
Nb	0.003 *	0.003 *	0.003 *	0.003 *
Ag	0.064 *	0.064 *	0.064 *	0.064 *
Cd	0.010	0.010	0.024	0.029 *
In	0.000	0.000	0.000	0.000
Sn	0.0333*	0.0333 *	0.0333 *	0.0333 *
Sb	0.007 *	0.006	0.007 *	0.007 *
Cs	0.008 *	0.003	0.006	0.003
Ta	0.000	0.000	0.000	0.000
W	0.002	0.003 *	0.003 *	0.002
Re	0.000	0.000	0.000	0.000

* All the values are the average of three replicates (*n* = 3), and the mean with the same letter (superscript) in the columns are not significantly different (*p* < 0.05)—(Tukey’s test).

**Table 4 plants-10-01426-t004:** The effect of mineral fertilisers on the mechanical composition of irrigated soil in the Kibray district.

Treatments	Fraction (%)							Physical Mud (%)
	1.0–0.25	0.25–0.1	0.1–0.05	0.05–0.01	0.01–0.005	0.005–0.001	<0.001	
Control	1.42	2.01	13.99	36.40	12.90 *	19.98 *	13.30	43.70 *
N_75_P_50_K_50_	2.02	3.36	16.10 *	35.11	11.10	18.20	14.11 *	43.41
N_125_P_100_K_100_	2.47	3.54 *	14.52	36.72 *	11.22	17.68	13.85	42.75
N_100_P_75_K_75_ + B3Zn6Fe6	3.78 *	3.14	15.00	36.77 *	9.70	17.51	14.10 *	41.31

* All the values are the average of three replicates (*n* = 3), and the mean with the same letter (superscript) in the columns are not significantly different (*p* < 0.05)—(Tukey’s test).

**Table 5 plants-10-01426-t005:** The effect of mineral fertilisers on agrochemical properties of irrigated soil in Kibray district.

Treatments	Active Phosphorus and Potassium, mg/kg		N-NO_3_, mg/kg	Total (%)		N (%)	Organic Carbon (%)	Organic Matter (%)	C/N
	P_2_O_5_	K_2_O		P_2_O_5_	K_2_O				
Control	34.85	238.88	12.01	0.21	0.80	0.098 *	0.9270	1.60	9.3
N_75_P_50_K_50_	40.02	245.88	12.57	0.25	0.84	0.098 *	0.9628	1.65	9.7
N_100_P_75_K_75_	44.60	249.47	25.58	0.27	0.95	0.097	0.9686	1.65	9.8
N_125_P_100_K_100_ + B_3_Zn_6_Fe_6_	48.9 *	251.44 *	37.98 *	0.29 *	0.98 *	0.098 *	0.9944*	1.70 *	10.1 *

* All the values are the average of three replicates (*n* = 3), and the mean with the same letter (superscript) in the columns are not significantly different (*p* < 0.05)—(Tukey’s test).

**Table 6 plants-10-01426-t006:** The effect of mineral fertilisers on chemical properties of irrigated soil in the Kibray district.

Treatments	CO_2_ (%)	Alkalinity									
		Total HCO_3_ (%)	Total HCO_3_ mg/eq	Cl (%)	Cl mg/eq	SO_4_ (%)	SO_4_ mg/eq	Ca (%)	Ca mg/eq	Mg (%)	Mg mg/eq
Control	8.20	0.02	0.36 *	0.05 *	0.20 *	1.07 *	0.50 *	0.18	19.01	8.20	0.02
N_75_P_50_K_50_	8.21	0.02	0.35	0.05 *	0.16	1.06	0.45	0.21 *	22.01 *	8.21	0.02
N_125_P_100_K_100_	8.25	0.02	0.34	0.04	0.15	0.67	0.42	0.18	21.14	8.25	0.02
N_100_P_75_K_75_ + B_3_Zn_6_Fe_6_	8.32 *	0.10 *	0.32	0.03	0.13	0.54	0.31	0.17	20.41	8.32 *	0.10 *

* All the values are the average of three replicates (*n* = 3), and the mean with the same letter (superscript) in the columns are not significantly different (*p* < 0.05)—(Tukey’s test).

**Table 7 plants-10-01426-t007:** The mechanical composition of irrigated soil in the Kibray district.

Land Use Types	Size of Mechanical Particle/mm							Physical Mud (%)	The Mechanical Content
	1–0.25	0.25–0.1	0.1–0.05	0.05–0.01	0.01–0.005	0.005–0.001	<0.001		
Cultivated land	4.35	6.89	10.99	36.18	12.64	14.99	13.96	41.59	Light sand

**Table 8 plants-10-01426-t008:** The agrochemical properties of irrigated soil in the Kibray district.

Land Use Types	Active Phosphorus and Potassium, mg/kg		N-NO_3_, mg/kg	Total (%)		N (%)	Organic Matter (%)	C (%)	C/N
	P_2_O_5_	K_2_O		P_2_O_5_	K_2_O				
Cultivated land	33.0	481.60	95.10	0.170	0.69	0.091	1.656	0.960	10.5

**Table 9 plants-10-01426-t009:** The chemical properties of irrigated soil in the Kibray district.

Land Use Types	CO_2_%	Alkalinity		Cl		SO_4_		Ca		Mg	
		Total HCO_3_ (%)	Total HCO_3_, м.экв	(%)	mg/eq	(%)	mg/eq	(%)	mg/eq	(%)	mg/eq
Cultivated	5.41	0.023	0.08	0.056	0.20	0.080	0.50	0.230	11.48	0.07	5.73

## Data Availability

All data, tables, figures, and results in paper are our own and original.

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
