# Peer review of "The Chemical Element Composition of Turmeric Grown in Soil–Climate Conditions of Tashkent Region, Uzbekistan"

_plants, 2021, doi:10.3390/plants10071426_

Round 1

Reviewer 1 Report

The manuscript entitled “The chemical element composition of turmeric grown in 2 soil-climate conditions of Tashkent Region, Uzbekistan” is interesting and enters within the line of the journal, however, there are some queries that should be considered before acceptance of this manuscript.

Comments as follows:

Major comments: Why the study is significant, should be added at the end of the abstract. Authors should add a paragraph in the introduction about How soil-climate conditions affect the chemical element composition of turmeric? The conclusion section is missing. Values presented in the table should be marked by decimal, not by comma and standard error or deviation should be added. Figure two not fully visualized.  

Minor comments:

  1. Line 19, Lysimeter write as lysimeter
  1. Line 40: Zingiberaceae should be italic.
  2. Line 40, Curcuma longa should be itelics
  3. Line 42: Curcuma longa should be italic.
  4. Line 72, remove the ‘rate’ as it is repeated twice in sentence.
  5. Line 75, remove the content word as it is using three times in sentence. As sentence should be like “significantly increased rhizome K, Ca, P and Mg content”.
  6. Line 76, use comma after Whereas…
  7. Line 81: “played” changed to “play”.
  8. Line 81, sentence should be finished with “different plants”.
  9. Line 101, The sentence should be rectify, like ‘enhanced the micronutrients like Fe, Mn, Zn, Cu, Cr and Si contents in turmeric rhizome as compared to control.
  10. Line 110, The sentence should be rewrite like this, Data regarding the ultra microelements content in turmeric rhizome showed non-significant difference in between all the treatment and control. In majority of the ultra-micronutrients (Li, Be,V. Co, Ni, Ga, Ge, Nb, Ag, Cd, Sn, Sb, Cs and W content), control showed higher value of rhizome nutrients than treatments (Table 3).
  11. Line 112, The sentence should be rewrite likewise, While, micronutrients like In, Ta and Re were absent in turmeric rhizome in all the treatments along with control.
  12. Line 120, give space in between “increased and mechanical”
  13. Comment:Line 123: to compared ”should be removed.
  14. Line 126 and 127, the sentence should be like as, “active P, K N content along with total P, K content and organic matter”.
  15. Line 127, The sentence should be rewrite like, “The highest values of total P, K content and N content, organic matter, active phosphorus, and potassium were observed in soil with mineral fertilizer treatments. The application rate of NPK (125:100:100 kg/ha) and NPK+BZnFe (100:75:75:3:6:6 kg/ha) improved nutrient contents of soil as compared to control and NPK (75:50:50 kg/ha) treatments.”
  16. Line 134, Please correct this, “Several authors reported that the nutrient contents in soil were analyzed before and after plant cultivation.”
  17. Line 135, “”Some other studies of poultry”””””
  18. Line 137, Dinesh et al. [39] reported the chemical nutrient management in which total N content of soil under rainfed ginger (Zingiber officinale) was positively enhanced .
  19. Line 138-139: Zingiberofficinaleshould be italic.
  20. Line 139, Similarly, the N, P and K content in soil were increased by the NPK application rate (100: 60: 60 kg/h) [40].
  21. In table 4, it should be Fraction (%) and Physical mud (%)
  22. In table 5, Total (%), N(%), Organic carbon (%), Organic matter (%)
  23. In Table 6, CO2(%), HCO3(%)…….
  24. Line 169, “the maximum the urease’ should be “The maximum urease””
  25. Line 170, The NPK+BZnFe (100:75:75:3:6:6 kg/h) treatment increased significantly urease activity which were higher than control and other treatments.
  26. Line 180, Sentence starting with, The combination of…..
  27. Line 204, remove the comma from Singh et al’
  28. Line 210, “the macro and micronutrient content of turmeric rhizome”
  29. Line 218: Curcuma longa should be italic.
  30. Line 232: hydrogen peroxide and nitric acid should be written formula
  31. Line 234: Nitric acid should be written formula
  32. Line 281: Zingiberofficinale should be italic.
  33. Line 285: Zingiberofficinale should be italic.
  34. Line 288-289: Curcuma longa should be italic.
  35. Scientific name of plants in references should be italic.

Author Response

Thank you for your valuable comments. We, the authors have corrected and amended all suggestions. 

Reviewer 2 Report

Dear Authors,

This is an interesting paper, which shows many detailed results on plant nutritional and mineral composition as well as soil condition after fertilization treatment of turmeric experiment. However, there are many details missing regarding the development of the paper that are needed to fully understand and interpret the results. In addition, the English language and the overall writing style should be considerably improved.

For instance, there are repetitions in several parts of the paper that are unnecessary. E.g., you repeat twice that the experiment took place at the Institute, which is relevant, but unnecessary to repeat twice in the abstract. This is only an example, there are several places in the text where a reordering of sentences and changes in structure would greatly benefit the readability and understandability of the text.

In addition, much information is missing, particularly regarding material and methods, which is absolutely needed to facilitate the understanding of the experiment and the evaluation of the experiment and its results.

Some recommendations:

Introduction

Interesting facts, but lack of consistence and cohesion, try to organize the introduction around 3-4 most important points, avoid repetitions and improve the English and the overall readability. Why is it important to add fertilizer to turmeric? In your region/in the World?

The last paragraph should explain very clearly why you conducted this study and why it is important, however, right now it is very unclear.

Results and discussion

Without a clearer description of the methods, what you did and why, it is very difficult to evaluate the results and the conclusions, as well as the importance of your results. Try to create a better link between your work and results from studies in the literature.

Materials and methods

The data analysis description is fuzzy. You mention a randomized block design, but this is never described, from what I see. Furthermore, you mention lysimeters, but I cannot find a sound description of those either.

Additionally, in the tables you mention that all means are calculated from three replicates, yet in the main text, you mention five replicates. What am I missing?

You mention ANOVA, but we don’t know which factors you take into account. I cannot see how the randomized blocking and the blocks are taken into account either.

Tables

This is interesting information, however, apart from the incongruence between 3 or 5 replicates (?), some measure about the variability is needed, for instance express for each treatment the mean value of (3? 5?) replicates, plus/minus 1 SE (standard error). This would provide the reader with a better insight on the results and the possible mechanisms acting behind.

Difficult to evaluate the soundness of your conclusions derived from those results without more details and additional information, plus some insight into the variability.

Figures

Again, this is interesting and valuable information, however, in addition to the comments above, fully applicable here, the type of graph should be changed. You do not have a gradient over a given mineral/nutritional element in the x-axis, but four different treatments. Thus, consider changing the dots and lines, better suited for increasing values of a given variable, by bars with some form of variability; again +/- 1 SE would be appropriate.

I encourage you to keep working to produce a better version of this manuscript for this or another journal.

Very best regards,

Teresa

Author Response

Comment: Extensive editing of English language and style required
Reply: The English language has been improved.

Comment: This is an interesting paper, which shows many detailed results on plant nutritional and mineral composition as well as soil condition after fertilization treatment of turmeric experiment. However, there are many details missing regarding the development of the paper that are needed to fully understand and interpret the results. In addition, the English language and the overall writing style should be considerably improved.

Reply: All the necessary missing details have been incorporated inside the text and the English language has been improved.

Comment: For instance, there are repetitions in several parts of the paper that are unnecessary. E.g., you repeat twice that the experiment took place at the Institute, which is relevant, but unnecessary to repeat twice in the abstract. This is only an example, there are several places in the text where a reordering of sentences and changes in structure would greatly benefit the readability and understandability of the text.

Reply: Repetitions have been removed from the text.

Comments: In addition, much information is missing, particularly regarding material and methods, which is absolutely needed to facilitate the understanding of the experiment and the evaluation of the experiment and its results.

Reply: the necessary missing details have been incorporated inside the methodology part.

Comment: Interesting facts, but lack of consistency and cohesion, try to organize the introduction around 3-4 most important points, avoid repetitions and improve the English and the overall readability. Why is it important to add fertilizer to turmeric? In your region/in the World?

Reply: All the repetitions have been removed and the introductory part has been modified with improving the language. In Uzbekistan especially in our Tashkent Region, the soil is very poor in terms of nutrients for turmeric production so extra fertilizer is much needed. As turmeric is log duration crop and it needs extra fertilizer for good yield and better quality so a high dose of nutrients with NPK fertilizer is added.

Comment: The last paragraph should explain very clearly why you conducted this study and why it is important, however, right now it is very unclear.

Reply: It is incorporated inside the text.

Comment: Without a clearer description of the methods, what you did and why, it is very difficult to evaluate the results and the conclusions, as well as the importance of your results. Try to create a better link between your work and results from studies in the literature.

Reply: Methodology has been described and incorporated with the results accordingly.

Comment: The data analysis description is fuzzy. You mention a randomized block design, but this is never described, from what I see. Furthermore, you mention lysimeters, but I cannot find a sound description of those either.

Reply: I have corrected the lysimeter as it was microplots.

Comments: Additionally, in the tables, you mention that all means are calculated from three replicates, yet in the main text, you mention five replicates. What am I missing?

Reply: The experiment has been conducted in three replicates, it was a typological error as we have rectified inside the text.

Comment: You mention ANOVA, but we don’t know which factors you take into account. I cannot see how the randomized blocking and the blocks are taken into account either.

Reply: Tukey test has been applied for the analysis and tables have been modified accordingly.

Comment: This is interesting information, however, apart from the incongruence between 3 or 5 replicates (?), some measure about the variability is needed, for instance, express for each treatment the mean value of (3? 5?) replicates, plus/minus 1 SE (standard error). This would provide the reader with a better insight into the results and the possible mechanisms acting behind them.

Reply: Three replications have been taken for the study.

Comments: Difficult to evaluate the soundness of your conclusions derived from those results without more details and additional information, plus some insight into the variability.

Reply: The conclusion part has been improved accordingly.

Comments: Again, this is interesting and valuable information, however, in addition to the comments above, fully applicable here, the type of graph should be changed. You do not have a gradient over a given mineral/nutritional element in the x-axis, but four different treatments. Thus, consider changing the dots and lines, better suited for increasing values of a given variable, by bars with some form of variability; again +/- 1 SE would be appropriate.

Reply: Graphs have been modified.

The reviewer comments are reasonable, and we have corrected the MS in accordance with the comments and suggestions. A thorough internal review was performed in the whole MS, changes highlighted in Track Change Format supplied MS. We are thankful to the learned reviewer for giving critical insights, leading to substantial improvement in the manuscript, we hope the response meets the reviewer approval.

Reviewer 3 Report

The work presented in this paper deals with the effect of fertilization on the  concentration level of 25 elements in the rhizome of turmeric and on some physical (size fractions, exchangeable cations), chemical (N, P, K content, organic matter) and biological soil properties (enzyme activities).

This data collection represents a great amount of work but it is difficult to understand the logics behind the fertilization experiments.  The N100P75K75 treatment T4 with B, Zn and Fe is not matched with an equivalent N100P75K75 treatment without B, Zn and Fe.

For this reason I cannot agree to the conclusions of the authors attributing all the observed effects to the B, Zn and Fe application.  The experimental design cannot permit to reach such a conclusion.

In addition the tables are not clearly organized. I do not understand why Tables 7, 8 and 9 are not merged respectively to Tables 4, 5 and 6. In the Figures 1, 2 and 3, the labels on the x axis are lacking.

Author Response

Comment: Moderate English change required
Reply: The English language has been improved.

Comment: This data collection represents a great amount of work but it is difficult to understand the logic behind the fertilization experiments. The N100P75K75 treatment T4 with B, Zn and Fe is not matched with an equivalent N100P75K75 treatment without B, Zn and Fe.

Reply: The treatment T4 and T3 are totally different as T3 have N125P100K100kg/ha and T4 have N100P75K75+B3Zn6Fe6kg/ha concentrations of the nutrients. In Uzbekistan, it is the first time cultivating turmeric crop so for the production of a good turmeric crop we have designed this fertilization experiment to achieve good quality and high yield and highly in term of nutrients in turmeric rhizome and soil. As in Uzbekistan soil is very poor.

Comment: For this reason, I cannot agree with the conclusions of the authors attributing all the observed effects to the B, Zn and Fe application. The experimental design cannot permit to reach such a conclusion.

Reply: As per our results, the T4 (N100P75K75+B3Zn6Fe6kg/ha) significantly increased the essential nutrients and enzymatic activities in the soil as well as macro and micronutrient contents in turmeric rhizome as compared to control.

Comment: In addition, the tables are not clearly organized. I do not understand why Tables 7, 8 and 9 are not merged respectively to Tables 4, 5 and 6.

Reply: Tables 4, 5 and 6 contain data from the turmeric rhizome. Tables 7, 8 and 9 show the soil data. So, Soil data cannot be merged.

Comment: In Figures 1, 2 and 3, the labels on the x-axis are lacking and interpret the results.

Reply: All the necessary missing details have been incorporated inside the figure legends.

Comment: In addition, the English language and the overall writing style should be considerably improved.

Reply: All the necessary missing details have been incorporated inside the text and the English language has been improved.

The reviewer comments are reasonable, and we have corrected the MS in accordance with the comments and suggestions. A thorough internal review was performed in the whole MS, changes highlighted in Track Change Format supplied MS. We are thankful to the learned reviewer for giving critical insights, leading to substantial improvement in the manuscript, we hope the response meets the reviewer approval.

Round 2

Reviewer 3 Report

My initial reluctance to the soundness of the fertilization experiments choice persists.  But I recognize that all other aspects of this paper have been greatly improved.  And I accept that this work can be useful for the turmeric production in Uzbekistan.

Author Response

To Editor/Reviewer #3

Comment: English language and style are fine/minor spell check required
Reply: English language and spelling have been improved.

Comment: The captions of Figures 1, 2 and 3 need clarifications. Bars represent means ± S.D or S.E.?

Reply: Bars represent means ± S.D which have been improved in the caption.

Comment: And with “*” what do you mean? *p<0.05, **p<0.001, ***p<0.001 vs control (not treated), according to the HSD Tukey’s post hoc test conducted after one-way ANOVA? If yes, please add “vs control (T1)” in the caption.
Reply: The “*” represent the results are significant compared to control which is mentioned inside the caption.

The reviewer and editors comments are reasonable, and we have corrected the MS in accordance with the comments and suggestions. A thorough internal review was performed in the whole MS, changes highlighted in Track Change Format supplied MS. We are thankful to the reviewer for giving critical insights, leading to substantial improvement in the manuscript, we hope the response meets the reviewer and editor approval.
